# Secure Transmission of Terahertz Signals with Multiple Eavesdroppers

**DOI:** 10.3390/mi13081300

**Published:** 2022-08-12

**Authors:** Yuqian He, Lu Zhang, Shanyun Liu, Hongqi Zhang, Xianbin Yu

**Affiliations:** 1College of Information Science and Electronic Engineering, Zhejiang University, Hangzhou 310027, China; 2Zhejiang Lab, Hangzhou 310000, China

**Keywords:** THz communications, physical layer security, multiple eavesdroppers, beam scattering, artificial noise

## Abstract

The terahertz (THz) band is expected to become a key technology to meet the ever-increasing traffic demand for future 6G wireless communications, and a lot of efforts have been paid to develop its capacity. However, few studies have been concerned with the transmission security of such ultra-high-speed THz wireless links. In this paper, we comprehensively investigate the physical layer security (PLS) of a THz communication system in the presence of multiple eavesdroppers and beam scattering. The method of moments (MoM) was adopted so that the eavesdroppers’ channel influenced by the PEC can be characterized. To establish a secure link, the traditional beamforming and artificial noise (AN) beamforming were considered as transmission schemes for comparison. For both schemes, we analyzed their secrecy transmission probability (STP) and ergodic secrecy capacity (ESC) in non-colluding and colluding cases, respectively. Numerical results show that eavesdroppers can indeed degrade the secrecy performance by changing the size or the location of the PEC, while the AN beamforming technique can be an effective candidate to counterbalance this adverse effect.

## 1. Introduction

Wireless traffic volume has exponentially grown in recent years and wireless data rates exceeding 100 Gbit/s will be required in the coming decades [1]. As a result, new frequency spectra are demanded to fulfill the broad bandwidth requirements for future communication. Among others, the THz band (0.06–10 THz) is regarded as a promising candidate to enable ultra-fast and ultra-broadband data transmission [2,3,4,5]. Recently, THz wireless communication systems are under rapid development and many wireless transmissions exceeding 100 Gbit/s have already been demonstrated in laboratories and in field environments [6,7,8,9,10,11], which bring THz communication closer to reality. However, ultra-high-speed THz communications also pose major challenges to information security [12,13]. Once a malicious eavesdropper tries to intercept the signals, a vast amount of information will be leaked in the blink of an eye which is absolutely unacceptable, particularly in some sensitive fields such as the military and financial industry.

Security mechanisms exist at every layer of a network. Compared to the conventional upper-layer methods [14,15], physical-layer security (PLS) approaches [16,17,18,19,20] do not rely on the assumption that eavesdroppers have limited computational abilities and avoid distributing and managing secret keys [21,22,23,24]. In contrast to the broadcast nature of the microwave communication, highly directive THz waves are more prone to the blockage problem caused by the malicious eavesdropper [25,26]. Recently, researchers have comprehensively investigated the blocking effects of an illegal recipient and proposed a hybrid beamforming and reflecting scheme to eliminate the adverse effects [27,28]. In this environment, any eavesdropper intending to hide itself should control its size, otherwise, it may cast a detectable shadow and raise an alarm. Therefore, the performance of eavesdropping is restricted by the size of the illegal receiver. Alternatively, recent works have pointed out that an eavesdropper may put a tiny passive object instead of itself, like a metal cup or a mobile phone, inside the narrow beam to scatter THz electromagnetic waves [29,30]. By this mean, the bulky illegal receiver placed outside the THz beam can capture the information signal without raising an alarm, as a consequence. We note that the feasibility of this scheme has already been demonstrated in experiments in which the eavesdropper can even intercept a signal strength as good as that of the intended receiver. Nevertheless, all the aforementioned work using scatter (tiny passive object) only consider a single-eavesdropper scenario while a case with multiple eavesdroppers has not been investigated. The reflector in the narrow beam scattering THz waves to multiple eavesdroppers may bring a greater security threat to the THz communication system.

Compared to the single eavesdropper, multiple eavesdroppers can increase the occurrence of stronger attackers that are closer to the legitimate transmitter due to the random spatial distribution [31,32]. Additionally, multiple eavesdroppers may also combine their own observations and jointly process their received message, which will considerably degrade the secrecy performance [33,34,35]. From a practical point of view, multi-eavesdropper scenes will be widespread phenomenon in our future, since potential eavesdroppers in the ubiquitous Internet of Things (IoT) may be some curious legitimate devices belonging to different subsystems [36]. However, secrecy performance and secure transmission schemes in highly directive THz communication systems have not been yet analyzed in the presence of multiple eavesdroppers. Moreover, how to safeguard this point-to-point THz system against randomly located eavesdroppers is still unknown.

In this paper, we comprehensively investigated the secrecy performance of a highly directive THz communication link with multiple eavesdroppers. We established the received signal models with two different multi-antenna techniques, namely traditional beamforming and AN beamforming, as transmission schemes for comparison. We note that the received signal mode is affected by the fading channel, where both the large-scale and small-scale effects matter. We emulate the effect of perfect electric conductor (PEC) parameters on the received signal-to-noise ratio (SNR) of Eve in a multiple-eavesdropper environment. We derive the mathematical framework of the STP and ESC in both non-colluding and colluding cases, so that the secrecy performance of the THz wireless link can be characterized. The results show that Eves can successfully intercept a huge amount of information by changing some parameters, such as the density, size, and distance. As a countermeasure, Alice could consider the deployment of the AN beamforming technique to counterbalance the adverse effect of multiple eavesdroppers.

The rest of the paper is organized as follows. In Section 2, we introduce the system model in the presence of multiple eavesdroppers. In Section 3, we analyze the STP and ESC in non-colluding and colluding cases, respectively. In Section 4, we conduct simulation experiments and demonstrate the factors affecting the secrecy performance. In Section 5, we discuss how one may find the attackers. Finally, we give a brief conclusion in Section 6. Additionally, the important notations in this paper are listed in Table 1 to make this paper clearer.

## 2. System Model

In this section, we first propose a security model for the THz system, in which two transmission schemes, namely traditional beamforming and AN beamforming, are adopted to prevent being overheard by multiple eavesdroppers. Then, the details of the highly directive channel of Bob $h_{B}$ and the scatter channel of Eve $h_{E}$ are investigated, respectively.

### 2.1. Signal Model

As shown in Figure 1a, a transmitter (Alice) sends a highly directive THz wave to the receiver (Bob) in the presence of multiple eavesdroppers (Eves). A PEC on the origin *O* is put inside this narrow beam between Alice and Bob. When there is an incident beam, PEC will scatter the THz signal to Eves in all directions (see Appendix A). We note that the PEC is located at the very edge of the THz beam with only a sliver of THz wave so it will not cast a detectable shadow in the receiver Bob. Additionally, the PEC is a cylinder which has the advantage of being able to scatter light in all directions, giving an attacker more flexibility. We model the locations of multiple eavesdroppers by the homogeneous Poisson point process (PPP) Φ in a circle region of radius $R_{S}$ with a density $\lambda_{p}$, as shown in Figure 1b. The total number of Eves $N_{E}$ in PPP is a random variable but the average number can be determined by $\bar{N_{E}}=\pi R_{S}^{2}\lambda_{p}$. Due to the short transmission distance ($R_{S}<15$ m) in an indoor environment, all receivers are supposed to be in a high SNR regime. Alice has *N* antennas while Bob and all the Eves use only one antenna each for reception.

When traditional beamforming is adopted, the received symbols at Bob and *i*-th Eve are, respectively, given by: (1)$$\begin{array}{cc} y_{B} & =h_{B}x+n_{B}, \end{array}$$(2)$$\begin{array}{cc} y_{E_{i}} & =h_{E_{i}}x+n_{E},\;i=1,2,\cdot \cdot \cdot ,N_{E}, \end{array}$$
where $h_{B}$ and $h_{E_{i}}$ are both 1×N vectors denoting the channel between Alice and Bob and between Alice and the *i*-th Eve, respectively; $N_{E}$ is the total number of eavesdroppers; $x=pu_{x}$ is the transmitted signal containing the beamforming vector p and signal $u_{x}$ with useful information; $n_{B}$ and $n_{E}$ are i.i.d. additive white Gaussian noise with $n∼\mathrm{CN}(0,\sigma_{n}^{2})$. We assume that both Alice and Bob only know the CSI of $h_{B}$, while Eve knows both $h_{B}$ and $h_{E_{i}}$ perfectly, which is a more rigorous scenario for the security issue [37,38,39].

With the introduction of AN beamforming, the transmitted THz signal x can be carefully designed as: x=s+w. The information signal $s=pu_{s}$, where the N×1 beamforming vector $p=h_{B}^{†}/\left|\right|h_{B}\left|\right|$ and signal $u_{s}$ with a variance of $\sigma_{u_{s}}^{2}$. The AN w=Zv, where the N×(N−1) matrix Z is the null space of vector $h_{B}$ so that $h_{B}Z=0$ while $h_{E}Z\neq 0$ and noise vector v contains (N−1) random noise elements with a variance of $\sigma_{v}^{2}$. Consequently, the received signals of Bob and *i*-th Eve are, respectively, given by: (3)$$\begin{array}{cc} y_{B} & =h_{B}\left(s+w\right)+n_{B}=h_{B}pu_{s}+n_{B}, \end{array}$$(4)$$\begin{array}{cc} y_{E_{i}} & =h_{E_{i}}\left(s+w\right)+n_{E}=h_{E_{i}}pu_{s}+h_{E_{i}}Zv+n_{E}. \end{array}$$

The AN w passes through the channel $h_{E_{i}}$ and finally develops into the additional noise $h_{E_{i}}w$. We stress that, despite the AN, the w on Alice’s side is sent to both the *i*-th Eve and Bob, whereas on the receiving side, the AN only deteriorates the *i*-th Eve without impacting Bob. As we can see, there is additional noise $h_{E_{i}}Zv$ on Equation (Equation 4) while there is no extra term on Equation (Equation 3).

The total transmitter power $P=E\left[x^{†}x\right]=\sigma_{u_{s}}^{2}+\left(N-1\right)\sigma_{v}^{2}$, where ${\left(\cdot \right)}^{†}$ denotes the conjugate transpose. We define η as the fraction of $\sigma_{u_{s}}^{2}$ to the total transmit power *P*. When η=1, the AN beamforming is equivalent to traditional beamforming as the information signal is transmitted with the full power *P*. We note that η is an important design parameter that can optimize the secrecy performance.

### 2.2. Highly Directive Channel

The channel model of Bob $h_{B}$ can be obtained as:(5)$$\begin{array}{c} h_{B}=l_{B}s_{B}, \end{array}$$
where $l_{B}$ is the large-scale factor denoting the fixed pass loss and $s_{B}$ is the small-scale random vector containing *N* elements. The $l_{B}$ influenced by the free space pass loss (FSPL) and highly directive antennas is given by:(6)$$\begin{array}{c} l_{B}=\frac{\lambda \sqrt{G_{t}G_{r}}}{4\pi d_{1}}, \end{array}$$
where $G_{t}$ and $G_{r}$, respectively, represent the antenna gains of Alice and Bob, and λ stands for the wavelength, and $d_{1}$ is the distance between Alice and Bob.

Unlike the conventional channel on the microwave band where the small-scale fading follows normal distribution, $s_{B}$ on the THz band is usually represented by Nakagami-m distribution with the *i*-th element $s_{B_{i}}∼Nakagami\left(m,1\right)$, which has recently been proven by experiments [40,41]. Finally, according to Equation (Equation 3), the signal-to-noise ratio (SNR) of Bob is given by:(7)$$\begin{array}{c} \mathrm{SNR}{}_{B}=S_{B}\frac{L_{B}P\eta }{\sigma_{n}^{2}}, \end{array}$$
where $S_{B}∼Gamma\left(mN,m\right)$ and $L_{B}=l_{B}^{2}$ are given by Equation (Equation 6).

### 2.3. Scatter Channel

The scatter channel of Eve $h_{E}$ is given by:(8)$$\begin{array}{c} h_{E_{i}}=l_{E_{i}}s_{E_{i}}, \end{array}$$
where $l_{{}_{i}}$ and $s_{E_{i}}$ are, respectively, the large-scale factor and small-scale random vector of *i*-th Eve. The $l_{E}$ and $s_{E}$ are totally different from $l_{B}$ and $s_{B}$ owing to the PEC between Alice and Bob. The PEC between Alice and Bob is a kind of material with infinite conductivity and zero electric field inside. When the incident field $E_{i}$ strikes the surface of PEC, it provokes a surface current $J_{Z}$ that generates a scattered field $E_{s}$ and total reflection occurs. By adopting the method of moments (MoM) [42], the scatter field $E_{s}$ around the PEC at *i*-th Eve is given by (see Appendix A):(9)$$\begin{array}{cc} E_{s} & =\frac{-k\eta_{0}}{4\pi }\sqrt{\frac{\eta_{0}PG_{t}}{kd_{2_{i}}}}exp\left\{-j\left(kd_{2_{i}}-\frac{\pi }{4}\right)\right\}C^{T}A^{-1}D, \end{array}$$
where *k* is the wave number, $\eta_{0}≃377\;\Omega$ is the intrinsic impedance of free space, $d_{2_{i}}$ is the distance between the PEC and *i*-th Eve and the matrices C,A,D are determined by the shape, size, and location of the PEC. Here, we assume that the PEC is a cylinder with sufficient height. As such, we can denote the scattering coefficient $K\left(a,d_{3}\right)=C^{T}A^{-1}D$, where *a* is the radius of PEC and $d_{3}$ is the distance between Alice and PEC. Therefore, the $l_{E_{i}}$ can be derived as:(10)$$\begin{array}{c} l_{E_{i}}=\sqrt{\frac{|E_{s}{|}^{2}}{2\eta_{0}}\frac{G_{r}\lambda^{2}}{4\pi P}}=\frac{\eta_{0}\lambda K\left(a,d_{3}\right)}{8\pi }\sqrt{\frac{kG_{t}G_{r}}{2\pi d_{2_{i}}}}, \end{array}$$
where we assume that Bob and all Eves have the same antenna gain $G_{r}$.

The scattering coefficient *K* is influenced by *a* and $d_{3}$. As shown in Figure 2, the THz wave nearly scatters uniformly around the PEC center ($d_{2}\gg \lambda$, [42]) and the scattered field gradually fades along as it becomes farther away from the center. The scattering coefficient *K* increases with radius *a* and decreases with $d_{3}$, as we can see since the color in Figure 2b is deeper than that in Figure 2a.

Unlike the main channel wherein a direct line-of-sight (LOS) link exists between Alice and Bob, Eve indirectly receives the signal information from non-line-of-sight (NLOS) transmission. Many rays will scatter from PEC and finally converge on Eve’s side as each point on the surface of PEC can generate an electromagnetic field. As such, a tiny move of PEC or Eve may tremendously change the received signal strength. Therefore, we assume $s_{E_{i}}∼Nakagami\left(1,1\right)$, which is also a *Rayleigh* distribution. Based on Equation (Equation 4), the SNR of *i*-th Eve is given by:(11)$$\begin{array}{c} \mathrm{SNR}{}_{E_{i}}=\frac{S_{E_{i}}L_{E_{i}}P\eta }{\frac{AL_{E_{i}}P\left(1-\eta \right)}{N_{A}-1}+\sigma_{n}^{2}}\overset{\left(a\right)}{\leq }\phi S_{equal_{i}}, \end{array}$$
where A∼Gamma(N−1), $S_{E_{i}}∼Exp\left(1\right)$, $L_{E_{i}}=l_{E_{i}}^{2}$, the PDF of random variable $S_{equal_{i}}$ is given by $f_{S_{equal}}\left(x\right)=\frac{N-1}{{\left(1+x\right)}^{N}}$ and $\phi =\frac{\eta (N-1)}{1-\eta }$, (a) holds for considering the worst-case situation where the normalized noise $\sigma_{n}$ are arbitrarily small. Note that this approach was also taken in [16,35,37].

## 3. Secrecy Performance

In this section, we introduce STP and ECS which are both secrecy performance metrics. Then, we analyze the secrecy performance with and without AN in both non-colluding and colluding cases.

### 3.1. Performance Metrics

In the non-colluding case, the eavesdropper individually overhears the communication between Alice and Bob without any centralized processing. Therefore, the SNR of multiple eavesdroppers is given by SNR${}_{E}$ = max (SNR${}_{E_{i}}$), where SNR${}_{E_{i}}$ is defined in Equation (Equation 11). Whereas, in the colluding case, $N_{E}$ Eves are capable of sending the information to a central data processing unit (CDPU) and jointly process their received information as shown in Figure 1a. Thus, the SNR of multiple eavesdroppers is given by SNR${}_{E}$ = $\sum_{i=1}^{N_{E}}$ SNR${}_{E_{i}}$. We adopt the following metrics to evaluate the secrecy performance of the proposed system.

*Secure transmission probability* (STP): STP is defined as a complementary element of secrecy outage probability (SOP) [31]. A supremum of the secrecy transmission rate *R* is determined by the difference of the main channel capacity $C_{B}=log\left(1+\mathrm{SNR}{}_{B}\right)$ and the wiretap channel capacity $C_{E}=log\left(1+\mathrm{SNR}{}_{E}\right)$. If secrecy transmission rates *R* are less than this supremum $C_{S}=C_{B}-C_{E}$, a secure transmission can be realized, otherwise, a secrecy outage occurs. The STP in non-colluding and colluding cases are, respectively, defined as: (12)$$\begin{array}{cc} P(C_{S}>R) & =\prod_{E_{i}\in \Phi }P\left(\frac{1+\mathrm{SNR}{}_{B}}{1+\mathrm{SNR}{}_{E_{i}}}>2^{R}\right), \end{array}$$(13)$$\begin{array}{cc} P(C_{S}>R) & =P(\frac{1+\mathrm{SNR}{}_{B}}{1+\sum \mathrm{SNR}{}_{E_{i}}}>2^{R}). \end{array}$$

*Ergodic secrecy capacity* (ESC): ESC is defined as the average transmission rate of the confidential message, which is formulated as:(14)$$\begin{array}{c} {\bar{C}}_{S}=E\left[C_{S}\right]=\int_{0}^{\infty }P\left(C_{S}>R\right)dR. \end{array}$$

In practice, ECS is used to describe the fast fading channel while STP for a slow fading channel. However, the numerical value of ECS is still determined by the STP. As long as we obtain the STP, the ESC can be simply calculated by its integration.

### 3.2. Non-Colluding Eavesdroppers

In a non-colluding eavesdroppers scenario, we investigated the STP for traditional beamforming (η=1) and AN beamforming (η≠1). When traditional beamforming is adopted, we derived the exact value of STP, whereas AN is introduced, and we calculated the lower bound of STP which is a rigorous assumption and common practice [16,37]. We denote the STP for traditional beamforming as $P_{1}$ and for AN beamforming as $P_{2}$, respectively. Based on Equation (Equation 12), $P_{1}$ is given by:(15)$$\begin{array}{c} P_{1}=\prod_{E_{i}\in \Phi }P\left(\frac{1+\mathrm{SNR}{}_{B}}{1+\mathrm{SNR}{}_{E_{i}}}>2^{R}\right)\overset{\left(b\right)}{=}E_{S_{B}}\left\{exp\left(-2\pi \lambda_{p}\int_{0}^{R_{S}}P\left(S_{E}>\frac{S_{B}L_{B}}{2^{R}L_{E}}\right)\rho d\rho \right)\right\}, \end{array}$$
where (b) holds for ${\mathrm{SNR}}_{B}\gg 1$, ${\mathrm{SNR}}_{E}\gg 1$ and the *probability generating functional lemma* (PGFL, ref. [43]) over PPP.

By denoting $u=kG_{t}G_{r}{\left(\eta_{0}K\lambda \right)}^{2}/128\pi^{3}$, we have $L_{E}=u\frac{1}{d_{2}}$. As $S_{E}∼E\left(1\right)$, the Equation (Equation 15) can finally be derived as:(16)$$\begin{array}{cc} P_{1}= & E_{S_{B}}\left\{exp\left(2\pi \lambda_{p}\left(\left(vR_{S}+v^{2}\right)e^{-\frac{R_{S}}{v}}-v^{2}\right)\right)\right\}, \end{array}$$
where $v=u2^{R}/S_{B}L_{B}$.

Similarly to the calculation of $P_{1}$ and by denoting $\beta =\frac{2^{R}\sigma_{n}^{2}}{PL_{B}}$, $P_{2}\left(C_{S}>R\right)$ is given by:(17)$$\begin{array}{c} P_{2}=\prod_{E_{i}\in \Phi }P\left(\frac{1+\mathrm{SNR}{}_{B}}{1+\mathrm{SNR}{}_{E_{i}}}>2^{R}\right)\overset{\left(c\right)}{=}E_{S_{B}}\left\{exp\left(\frac{-\pi R_{S}^{2}\lambda_{p}}{{\left(1+\frac{S_{B}\eta -\beta }{\beta \phi }\right)}^{N-1}}\right)\right\}, \end{array}$$
where (c) holds for ${\mathrm{SNR}}_{B}\gg 1$ and the PGFL over PPP.

### 3.3. Colluding Eavesdroppers

In colluding case, we denote the STP without AN as $P_{3}$ and with AN as $P_{4}$, respectively. The STP $P_{3}\left(C_{S}>R\right)$ is given by:(18)$$\begin{array}{c} P_{3}=P\left(\frac{1+\mathrm{SNR}{}_{B}}{1+\sum \mathrm{SNR}{}_{E_{i}}}>2^{R}\right)=P\left(S_{B}>\frac{2^{R}\sum_{E_{i}\in \Phi }S_{E_{i}}L_{E_{i}}}{L_{B}}\right). \end{array}$$

We let $I_{1}=\sum_{E_{i}\in \Phi }S_{E_{i}}L_{E_{i}}$ and thus $P_{3}$ can be modified as:(19)$$\begin{array}{c} P_{3}=\int_{0}^{\infty }P\left(S_{B}>p_{1}i\right)f_{I_{1}}\left(i\right)di\overset{\left(d\right)}{=}\sum_{b=0}^{mN-1}m_{b}p_{1}^{b}{\left(-1\right)}^{b}L^{\left(b\right)}\left\{f_{I_{1}}\left(i\right)\right\}\left(mp_{1}\right), \end{array}$$
where $f_{I_{1}}\left(i\right)$ is the probability density function (PDF) of $I_{1}$ and $p_{1}=2^{R}/L_{B}$, (d) holds for $S_{B}∼Gamma\left(mN,m\right)$ and the complementary cumulative distribution function (CCDF) of $S_{B}$ is given by $F_{S_{B}}^{c}=e^{-mx}\sum_{b\in B}m_{b}x^{b}$, where $m_{b}=\frac{m^{b}}{b!}$ and b∼(0,mN−1). The Laplace transformation $L\left\{f_{I_{1}}\left(i\right)\right\}\left(mp_{1}\right)$ of function $f_{I_{1}}\left(i\right)$ is given by:(20)$$\begin{array}{c} L\left\{f_{I}\right\}\left(p_{1}\right)=exp\left\{-2\pi \lambda_{p}p_{1}uR_{S}\right\}{\left(1+\frac{R_{S}}{p_{1}u}\right)}^{2\pi \lambda p_{1}^{2}u^{2}}. \end{array}$$

By adopting *Bruno’s formula* [44], we can obtain the *n*-degree derivation of $L\left\{f_{I_{1}}\left(i\right)\right\}\left(p_{1}\right)$ as:(21)$$\begin{array}{c} L^{\left(n\right)}\left\{f_{I_{1}}\right\}\left(p_{1}\right)=\sum \frac{n!}{b_{1}!\cdot \cdot b_{n}!}e^{f(p_{1})}\prod_{j=1}^{n}{\left(\frac{f^{\left(j\right)}\left(p_{1}\right)}{j!}\right)}^{b_{j}}, \end{array}$$
where the sum is over all the solutions $b_{1},\cdot \cdot \cdot ,b_{n}\geq 0$ to $b_{1}+2b_{2}+\cdot \cdot \cdot +nb_{n}=n$. By denoting $w=R_{S}/u,c_{1}=2\pi \lambda_{p},c_{2}=1+\frac{w}{p_{1}},c_{3}=\frac{1}{p_{1}+w}-\frac{1}{p_{1}},c_{4}=\frac{1}{p_{1}^{2}}-\frac{1}{{\left(p_{1}+w\right)}^{2}}$, $f(p_{1})$ and $f^{\left(j\right)}\left(p_{1}\right)$ are given by:
(22a)$$\begin{array}{cc} f(p_{1})= & c_{1}\left(p_{1}^{2}u^{2}lnc_{2}-p_{1}uR_{S}\right), \end{array}$$
(22b)$$\begin{array}{cc} f^{\left(1\right)}\left(p_{1}\right)= & c_{1}\left(2p_{1}u^{2}lnc_{2}+p_{1}^{2}u^{2}c_{3}-uR_{S}\right), \end{array}$$
(22c)$$\begin{array}{cc} f^{\left(2\right)}\left(p_{1}\right)= & c_{1}\left(2u^{2}lnc_{2}+4p_{1}u^{2}c_{3}+p_{1}^{2}u^{2}c_{4}\right), \end{array}$$
(22d)$$\begin{array}{c} f^{\left(j>2\right)}\left(p_{1}\right)=c_{1}u^{2}\sum_{kk=0}^{2}C_{2}^{kk}{\left(-1\right)}^{j-kk}\frac{j!}{\left(j-kk\right)}p_{1}^{2-kk}\left(\frac{1}{p_{1}^{j-kk}}-\frac{1}{{\left(p_{1}+w\right)}^{j-kk}}\right). \end{array}$$

When AN beamforming is introduced, $P_{4}\left(C_{S}>R\right)$ is given by:(23)$$\begin{array}{c} P_{4}=\int_{0}^{\infty }P\left(S_{B}>p_{2}i\right)f_{I_{2}}\left(i\right)di=P\left(S_{B}>\frac{\beta (1+\sum_{E_{i}\in \Phi }\phi S_{equal_{i}})}{\eta }\right). \end{array}$$

We let $I_{2}=1+\sum_{E_{i}\in \Phi }\phi S_{equal_{i}}$ and thus $P_{4}$ can be rewritten as:(24)$$\begin{array}{c} P_{4}=\int_{0}^{\infty }P\left(S_{B}>p_{2}i\right)f_{I_{2}}\left(i\right)di=\sum_{b=0}^{mN-1}m_{b}p_{2}^{b}{\left(-1\right)}^{b}L^{\left(b\right)}\left\{f_{I_{2}}\left(i\right)\right\}\left(mp_{2}\right), \end{array}$$
where $f_{I_{2}}\left(i\right)$ is the PDF of $I_{2}$ and $p_{2}=\frac{\beta }{\eta }$. As long as we obtain $L\left\{f_{I_{2}}\left(i\right)\right\}\left(mp_{2}\right)$, $P_{4}$ can be calculated. The Laplace transformation $L\left\{f_{I_{2}}\left(i\right)\right\}\left(p_{2}\right)$ is given by:(25)$$\begin{array}{c} L\left\{f_{I_{2}}\right\}\left(p_{2}\right)=exp\left\{-p_{2}-N_{E}+q_{1}q_{2}\right\}, \end{array}$$
where $q_{1}=exp\left(p_{2}\phi \right)E_{N}\left(p_{2}\phi \right)$, $E_{N}\left(x\right)=\int_{1}^{\infty }\frac{e^{-xt}}{t^{N}}dt$ is the *N*-degree exponential integral and $q_{2}=N_{E}\left(N-1\right)$. As such, the *n*-degree of $L\left\{f_{I_{2}}\left(i\right)\right\}\left(p_{2}\right)$ is given by:(26)$$\begin{array}{cc} L^{\left(n\right)}\left\{f_{I_{2}}\right\}\left(p_{2}\right) & =\sum \frac{n!}{b_{1}!\cdot \cdot b_{n}!}e^{g(p_{2})}\prod_{j=1}^{n}{\left(\frac{g^{\left(j\right)}\left(p_{2}\right)}{j!}\right)}^{b_{j}}, \end{array}$$
where $g(p_{2})$ and $g^{\left(j\right)}\left(p_{2}\right)$ are given by:
(27a)$$\begin{array}{c} g\left(p_{2}\right)=-p_{2}-N_{E}+q_{1}q_{2}, \end{array}$$
(27b)$$\begin{array}{c} g^{\left(1\right)}\left(p_{2}\right)=-1+q_{2}\phi e^{k\phi }\left(E_{N}-E_{N-1}\right), \end{array}$$
(27c)$$\begin{array}{c} g^{\left(j\geq 2\right)}\left(p_{2}\right)=q_{2}\phi^{j}e^{k\phi }\sum_{jj=0}^{j}C_{j}^{jj}{\left(-1\right)}^{jj}E_{N-jj}. \end{array}$$

## 4. Security Analysis

In what follows, we describe Eve’s strategies to degrade the secrecy performance with a PEC, and then we show the function of AN as a countermeasure to resist the multiple eavesdroppers. Meanwhile, power allocation as a significant parameter of AN beamforming is also analyzed. Table 1 shows the parameter settings.

### 4.1. Eve’s Attack

The intensity of Eves’ attack is affected by the spatial distribution. In Figure 3a, when we compare the blue line with the red and yellow line, respectively, we find that the covering radius $R_{S}$ has little effect on STP while density $\lambda_{p}$ significantly reduces the STP. However, in Figure 3b, both the value of $R_{S}$ and $\lambda_{p}$ have significant impacts on the STP. The reason is that the SNR of multiple eavesdroppers in the non-colluding case depends on the ‘nearest’ Eve which has the best channel quality while the SNR in the colluding case only depends on the total number. The parameter $R_{S}$ can barely increase the chance of the ‘nearest’ Eve as the THz transmit power quickly attenuates with the distance but indeed increases the total number of them. Therefore, from Eves’ perspective, they have to focus on ‘a better channel’ or ‘a better location’ rather than the total number in a non-colluding case, as we can see the STP of the case when $\bar{N_{E}}$ = 17 performs even better than the STP when $\bar{N_{E}}=7$ in Figure 3a.

In Figure 4, we use normalized secrecy capacity [30] to show the extent to which Eves reduce the secrecy capacity in non-colluding and colluding cases, respectively. It is shown that for $d_{2}$ = 20, the existence of Eves reduces the original capacity by 20% in non-colluding case and by nearly 40% in colluding case.

Eve can move the PEC closer to Alice to strengthen the attack. In Figure 5, we find that the ESC monotonically increases with $d_{3}$ (PEC) while decreasing with the $d_{1}$ (Bob). In addition, the parameters $d_{1}$ and $d_{3}$ may have interacted with each other. For example, for $d_{1}$ = 3, a unit increase in $d_{3}$ will give birth to the improved ESC by ΔESC = 1.45. For $d_{1}$ = 5, ΔESC becomes 1.95. That is to say, $d_{3}$($d_{1}$) may exhibit a different effect when the other factor changes. Furthermore, if PEC is located in the midpoint between Alice and Bob, the ESC will not change significantly with the increase in $d_{1}$, as we can see that the white line in Figure 5 nearly remains unchanged at ESC = 4.85.

Eve can increase the size of PEC to strengthen the attack. In Figure 6a, we find that the STP will decrease when the radius *a* rises from 20 mm to 40 mm, regardless of whether it is in the non-colluding case or in the colluding case. As shown in Figure 2, as *a* grows from 20 mm to 40 mm, the electromagnetic field around the PEC will be augmented and hence Eves obtains better signal quality. Additionally, we find that Eves benefit from increasing *a* to various degrees when the location of PEC $d_{3}$ changes. For $d_{3}$ = 5 m, as shown in Figure 6b, reducing *a* from 10 mm to 50 mm will lead to an ESC decrease of 36%. For $d_{3}$ = 1 m, however, reducing *a* from 10 to 50 decreases the ESC by 87% to nearly 0 which means that Eves can almost intercept all the information. Since being too near to Alice will increases the risk of being detected, Eve’s strategy is to select a proper size and optimal location in such a way she can obtain as good a signal strength as possible and hide herself simultaneously.

### 4.2. AN as a Countermeasure

We find that the AN beamforming can compensate for the detriment of multiple eavesdroppers. As shown in Figure 7, the increase in $\lambda_{p}$ causes an STP ($P(C_{S}\geq 0)$) reduction from 0.85 to 0.75 and 0.5 to 0.1, respectively. However, with the introduction of AN in the non-colluding case, the STP ($P(C_{S}\geq 0)$) rises to 0.95, leading to an improvement of nearly 27%. In the colluding case, the STP rises to 0.7, corresponding to an improvement of 600%. It is noteworthy that the detriment of multiple eavesdroppers in the colluding case is more than that in the non-colluding case. In the non-colluding case, for R>2, the STP with AN beamforming ($\lambda_{p}=0.02$) is higher than that with traditional beamforming ($\lambda_{p}=0.01$). However, in the colluding case, the situation is reversed for R>2 which means that colluding eavesdroppers cause greater damage to transmission security.

In Figure 8, we find that the optimal η depends on both density $\lambda_{p}$ and the number of antennas *N*. For $\lambda_{p}=0.01$ (blue and yellow line), the optimal η in the non-colluding and colluding cases is 0.28 and 0.22, respectively, which are larger than 0.21 and 0.15 for $\lambda_{p}=0.02$. More Eves around PEC signify stronger information attacks. Therefore, Alice must allocate more transmission power to AN to resist the adverse effect of the added Eves. Additionally, the optimal value of η increases with *N*. As shown in the inset, the optimal η are 0.34 and 0.27 for N=6 while 0.28 and 0.22 for N=2. We stress that only Bob benefits from the increase in antennas since the transmitter maximizes the signal strength to Bob and the signal power at Eves’ side remains unchanged.

In Figure 9, we find that the ESC decreases with the *f* while increasing with the *P* when η≠1. For a system without AN, the ESC will not be influenced by *P* since ${\mathrm{SNR}}_{B}$ and ${\mathrm{SNR}}_{E}$ benefit from them to the same extent, as shown by Equations (Equation 7) and (Equation 11). However, with the introduction of AN, *P* can no longer influence the supremum of ${\mathrm{SNR}}_{E}$ but still impacts ${\mathrm{SNR}}_{B}$. Additionally, we also find *P* and *f* cannot significantly change the optimal η. In Figure 9a,b, the optimal η varies in the ranges of 0.27∼0.31 and 0.26∼0.3 with standard deviations (STD) of 1.13 $\times \;10^{-2}$ and 1.14 $\times \;10^{-2}$, respectively, lending to a tiny change. We note that despite Figure 9 only showing a non-colluding case, the same rule can also be applied to the colluding scenario.

## 5. Discussion

In practice, the first step to guarantee transmission security is to determine whether attackers exist instead of determining how to resist attackers. Therefore, before using unique techniques (such as AN), we should adopt a specific measure to detect the existence of an attacker, otherwise, many resources will be wasted. Recent work in [30] can successfully distinguish the suspicious objects from the ordinary environment through measuring the incoming signal. Here, we consider the possibility of increasing the beam directivity or enlarging the aperture of the receiver to guarantee the security. In this paper, the diameter of the THz beam is larger than the aperture of the receiver. Thus, Eves can utilize the edge of the beam to realize an attack. However, if the receiver has the ability to capture all of the transmitted THz wave without any leakage, any eavesdroppers trying to put an object in the beam will cause an extensive power reduction on Bob’ side. In this case, if Eves still wants to implement an attack, she needs to either utilize the misalignment effect between Alice and Bob which may also induce a leakage or pretend to be irrelevant moving objects. Nevertheless, either way, Eves’ strategy to implement an attack would be significantly more complicated and harder to implement. Another purpose of increasing the directivity is to resist the interference. Transceivers on the same unlicensed bandwidth may have interacted with each other. Additionally, jammers can also take advantage of this large bandwidth in the THz band for interference [45]. Increasing their directivity gains can make irrelevant transceivers and jammers either less effective or need to increase their transmit power.

In some cases, Eves are not afraid of being found because they are intended to block the signal power of Bob (reduce the secrecy capacity at the same time). As a countermeasure, multiple IRS-assisted THz systems with opportunistic connectivity may be a choice since Alice can choose different ways to transmit the signal and design unique beamforming schemes to maximize the secrecy rate performance. Researchers have found that opportunistic connectivity [46] with well-designed beamforming schemes can significantly boost the secrecy rate performance and reduce blocking probability.

## 6. Conclusions

In this paper, we investigated the secure transmission of THz waves in the indoor environment against randomly distributed eavesdroppers. We established the PLS model for this THz communication system, where Bob’s channel is featured by a highly directive beam while Eve’s channel scatters THz waves. Particularly, we characterize both channels with stochastic small-scale fading in order to accommodate the random variation in practice such as scattering on aerosols or the movement of objects. The security performance of traditional beamforming and AN beamforming in both non-colluding and colluding cases are analyzed by deriving the STP and ESC. Based on our analysis, we reveal that Eves can indeed take different strategies to degrade the secrecy performance, for instance, by changing the size or the distance of the scatter and increasing the density. To deal with this issue, an AN beamforming technique with a well-designed power allocation can be an effective candidate to counterbalance this adverse effect. Our study can not only serve as an inspiration for eavesdropping scenes but also for a widespread network scenario. Future work may extend this point-to-point communication scene to an indoor THz wireless local area networks (WLANs) which seem more appealing.

## Figures and Tables

**Figure 1 micromachines-13-01300-f001:**
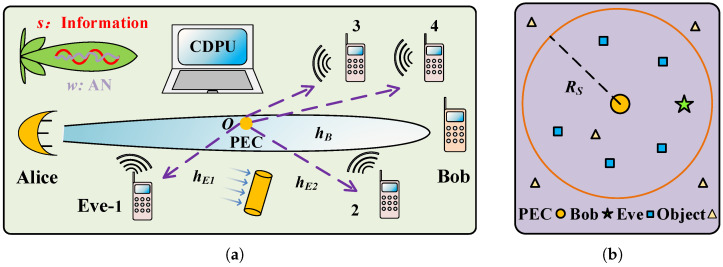
System model. (**a**) Alice transmits a highly directive THz signal x to Bob with or without AN w. A PEC (orange cylinder) located at the edge of beam can scatter the incident THz wave to Eves in all directions. (**b**) The spatial distribution of Eves is modeled as PPP in a circle region. The objects in this indoor scene can scatter THz signals.

**Figure 2 micromachines-13-01300-f002:**
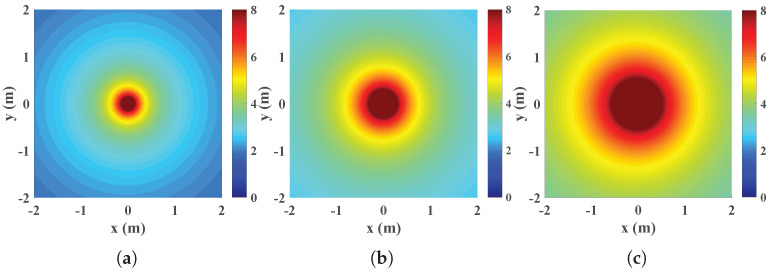
The scattered fields of PEC for (**a**) *a* = 20 mm, $d_{3}$ = 2 m (**b**) *a* = 40 mm, $d_{3}$ = 2 m (**c**) *a* = 40 mm, $d_{3}$ = 1.5 m. The maximum values were cut off at 8 since only a few values exceed it.

**Figure 3 micromachines-13-01300-f003:**
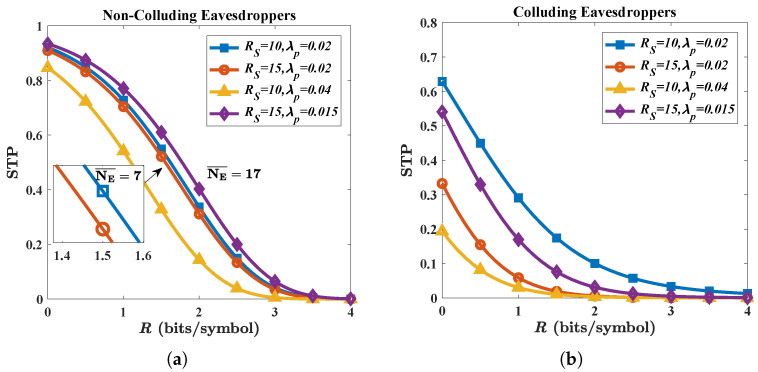
Secure transmission probability (STP) under different $R_{S}$ and $\lambda_{p}$ for (**a**) non-colluding case and (**b**) colluding case. Parameters are given by: *G* = 25 dBi; *N* = 5; *f* = 300 GHz; and *P* = −10 dBm.

**Figure 4 micromachines-13-01300-f004:**
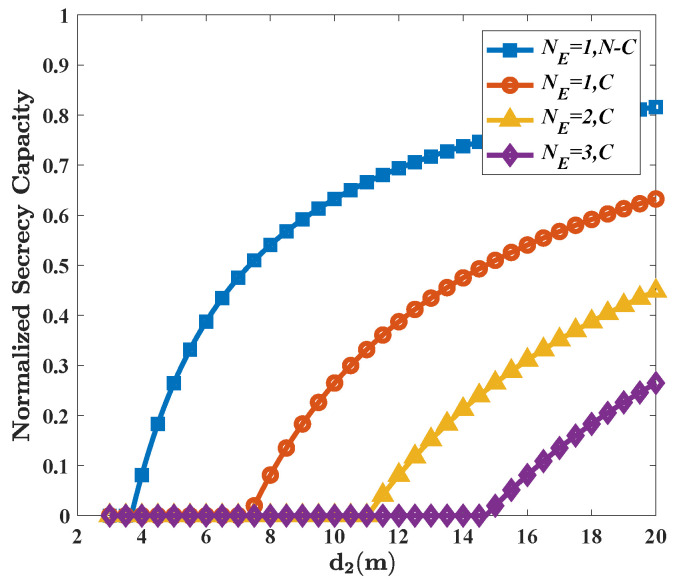
The normalized secrecy capacity as a function of $d_{2}$ in the non-colluding and colluding cases. Here, all the eavesdroppers have the same distance $d_{2}$ to the PEC and the channel fading is ignored. Other parameters are given by: *G* = 25 dBi; *f* = 300 GHz; *P* = −10 dBm; $R_{S}$ = 15 m; and $d_{3}$ = 1 m.

**Figure 5 micromachines-13-01300-f005:**
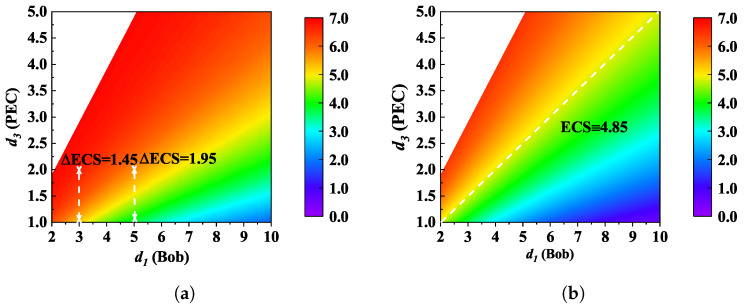
Ergodic secrecy capacity (ESC) as a function of $d_{1}$ (Alice–Bob) and $d_{3}$ (Alice–PEC) for (**a**) non-colluding eavesdroppers and (**b**) colluding eavesdroppers. Other parameters are given by: G=25 dBi; *N* = 3; *f* = 300 GHz; *P* = −10 dBm; $R_{S}$ = 15 m; and $\lambda_{p}$ = 0.015.

**Figure 6 micromachines-13-01300-f006:**
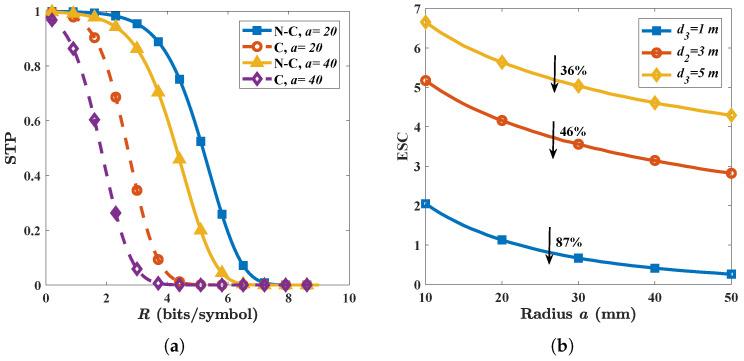
(**a**) Influence of radius *a* on STP. (**b**) ESC versus radius *a* under different PEC location $d_{3}$. The solid line describes the non-colluding case while the dashed line describes the colluding case. Other parameters are given by: *G* = 25 dBi; *N* = 3; *f* = 300 GHz; *P* = −10 dBm; $R_{S}$ = 15 m; and $\lambda_{p}$ = 0.04.

**Figure 7 micromachines-13-01300-f007:**
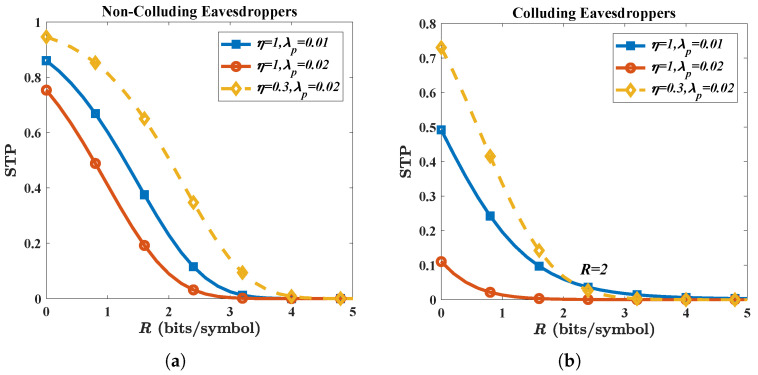
The benefit of AN on STP for (**a**) a non-colluding case; and a (**b**) colluding case. Other parameters are given by: *G* = 25 dBi, *N* = 3, *f* = 300 GHz, *P* = −10 dBm, $R_{S}$ = 15 m, η = 0.3.

**Figure 8 micromachines-13-01300-f008:**
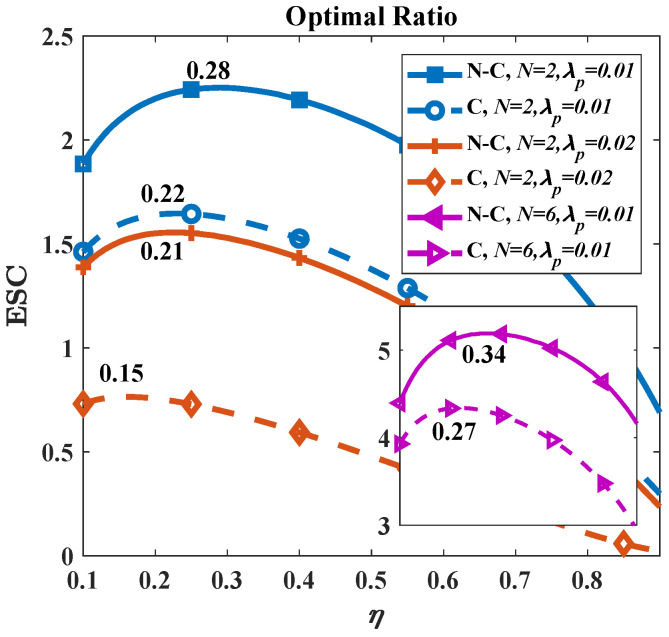
The optimal η under different $\lambda_{p}$ and *N*. The solid line describes non-colluding cases while the dashed line describes colluding cases. The main figure for *N* = 2 while the inset for *N* = 6. Other parameters are given by: *G* = 27 dBi; *f* = 300 GHz; *P* = −10 dBm; and $R_{S}$ = 15 m.

**Figure 9 micromachines-13-01300-f009:**
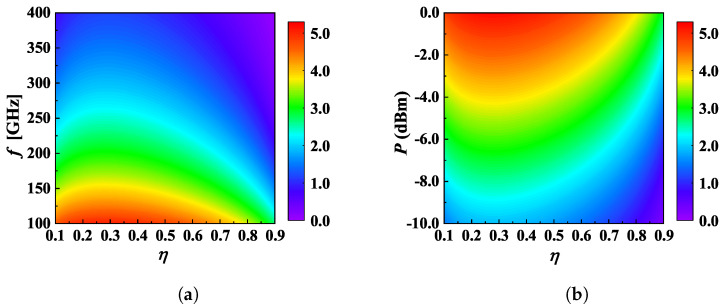
Secrecy performance in a non-colluding case. (**a**) The ECS as a function of η and *f* with P=−10 dBm; (**b**) The ECS as a function of η and *P* with *f* = 300 GHz. Other parameters are given by: *G* = 25 dBi, *N* = 3, $R_{S}$ = 15 m, $\lambda_{p}$ = 0.02.

**Table 1 micromachines-13-01300-t001:** Parameter settings.

Side	Symbol	Parameter Setting	Value
Alice	*P*	Transmitting power	−10 dBm
	$G_{t}$	Antenna gain	25/27 dBi
	*N*	Antenna number	Independent variable
	η	Power allocation ratio	Independent variable
Channel	$R_{S}$	Covering radius	10/15 m
	$\lambda_{p}$	Density of eavesdroppers	Independent variable
	$N_{E}$	Number of eavesdroppers	Independent variable
	*a*	Radius of cylinder	Independent variable
	$d_{2}$	Distance between Eve and PEC	Independent variable
	$d_{3}$	Distance between Alice and PEC	Independent variable
	*m*	Nakagami fading parameters	2
Bob	$d_{1}$	Distance between Alice and Bob	Independent variable
	$G_{r}$	Antenna gain	25/27 dBi
Other	*c*	Speed of light	3 × 10${}^{8}$ m/s
	*f*	Frequency	Independent variable
	$P_{N}$	Noise power	−75 dBm
	*W*	Bandwidth	50 GHz
	N-C	Non-colluding case	-
	C	Colluding case	-

## Data Availability

Not applicable.

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
