# Peer review of "Secure Transmission of Terahertz Signals with Multiple Eavesdroppers"

_micromachines, 2022, doi:10.3390/mi13081300_

Round 1

Reviewer 1 Report

The authors addressed the security concerns when multiple eavesdroppers are considered in a THz signal transmission. This paper is well written and seems technically sound. However, the following issues should be considered.

1. PEC is not defined.

2. Should cite recent articles that focus on similar issues and include a comparative discussion. See https://doi.org/10.1109/TVT.2022.3172763  

Reviewer 2 Report

1. Please refer to the 'Study exposes security vulnerabilities in terahertz data links; (https://phys.org/news/2018-10-exposes-vulnerabilities-terahertz-links.html), which discusses how placing objects at the very edge of a beam that is capable of scattering a tiny portion of the beam can be a cause. I suggest the authors discuss the More directional beam in THz in the article.

2. Authors have discussed SNR (which is good); I suggest the author discuss 'Normalized Secrecy Capacity,' a characterization of the maximum transmission rate at which the eavesdropper cannot decode any information.

3. PLS based on artificial noise (AN) and PLS channel coding are the most used techniques for achieving this channel security capacity. PLS based on artificial noise (AN) and PLS channel coding provides interference to eavesdroppers to reduce the channel capacity of eavesdropping channels and increase the channel security capacity. The PLS employing AN interferes with eavesdroppers to reduce the channel capacity of eavesdropping channels and increase the channel security capacity. The authors have discussed AN in the abstract but didn't mention it anywhere else in the paper as a study. Also, I suggest discussing Channel Coding.

4. THz of unlicensed bandwidth is also an issue of secrecy, and eavesdroppers' authors should include the discussion in the article.

5. I also suggest that authors include, what are the possible strategies researchers are discussing, and possible implementation? How effective are those techniques?

Round 2

Reviewer 1 Report

This paper can be accepted in its current form.